# Participants’ Experiences of the 2018–2019 Government Shutdown and Subsequent Supplemental Nutrition Assistance Program (SNAP) Benefit Disruption Can Inform Future Policy

**DOI:** 10.3390/nu12061867

**Published:** 2020-06-23

**Authors:** Wendi Gosliner, Wei-Ting Chen, Cathryn Johnson, Elsa Michelle Esparza, Natalie Price, Ken Hecht, Lorrene Ritchie

**Affiliations:** 1Nutrition Policy Institute, Division of Agriculture and Natural Resources, University of California, Berkeley, CA 94704, USA; kenhecht@ucanr.edu (K.H.); lritchie@ucanr.edu (L.R.); 2Stanford University School of Medicine, Stanford, CA 94305, USA; weiting.chen@stanford.edu; 3Division of Agriculture and Natural Resources, Cooperative Extension, University of California, Davis, CA 95618, USA; ckrjohnson@ucanr.edu (C.J.); nmprice@ucanr.edu (N.P.); 4School of Public Health, University of California, Berkeley, CA 94704, USA; eesparza@berkeley.edu

**Keywords:** Supplemental Nutrition Assistance Program, federal government shutdown, food insecurity, qualitative research, safety net, nutrition

## Abstract

The federal government shutdown from 22 December 2018 to 25 January 2019 created an unprecedented disruption in Supplemental Nutrition Assistance Program (SNAP) benefits. We conducted a cross-sectional qualitative study to begin to capture how the disruption affected food security and wellbeing among a small sample of California SNAP participants. We collected data from 26 low-income adults in four focus groups in four diverse California counties. We found that participants routinely struggle to secure an adequate and healthy diet in the context of high costs of living, the shutdown and benefit disruption added to participants’ stress and uncertainty and exacerbated food insecurity, and it diminished some participants’ faith in government. Participants reported that, while having additional benefits in January felt like a relief from typical end-of-month deprivation, the subsequent extended gap between benefit distributions and a lack of clarity about future benefits caused cascading effects as participants later had to divert money from other expenses to buy food and faced added uncertainty about future economic stability. Additionally, the shutdown highlighted challenges related to the availability, timing, and tone of communications between participants and SNAP agencies. Participants recommended that SNAP adjust benefit and eligibility levels to better address costs of living, improve customer service, and avoid future disruptions.

## 1. Introduction

The Supplemental Nutrition Assistance Program (SNAP, formerly known as food stamps), provides critical funds to low-income families to support food purchases, helping to alleviate poverty and food insecurity. In an average month in 2019, more than 35 million Americans participated in SNAP, at a total cost of approximately $60 billion dollars annually [1]. In 2018, SNAP lifted 3.2 million people out of poverty [2]; the poverty reducing benefits of SNAP are so large that, without the program, the child poverty rate in the U.S. would be 18% instead of 13% [3]. More than 67% of SNAP participants are families with children, almost 34% live in households with members who are elderly or have disabilities, and more than 43% of SNAP participants are in working families [4]. While SNAP reduces the overall prevalence of food insecurity by as much as 30%, more than 37 million Americans, including more than half of households participating in SNAP, struggle with food insecurity [5,6,7]. Rates of food insecurity among non-Hispanic Black (21.2%) and Hispanic (16.2%) households are substantially higher than for White, non-Hispanic (8.1%) households [7].

Even with SNAP, many low-income families struggle to meet their basic needs [8,9,10]. Multiple studies have reported that SNAP benefits are too low and benefit increases have not kept pace with rising costs [11,12,13,14,15]. Annual cost of living adjustments are made to SNAP benefit levels to account for national inflation in the cost of food; however, regional variations in food prices are not considered [16]. A report from Feeding America found that almost 26 million food-insecure individuals reside in areas where food costs are above the national average [17]. Additionally, studies have found that the Thrifty Food Plan (TFP), USDA’s estimate of a nutritionally adequate diet upon which SNAP benefit levels are based, is flawed as it does not account for the time-cost of preparing meals [14,18,19]. For most SNAP households, benefits do not last until the end of the month, and many participants experience hunger or are forced to sacrifice dietary quality when SNAP benefits are exhausted [20].

While SNAP provides numerous benefits to participants, persistent struggles with stigma have been documented [21]. People who work yet do not earn enough to be self-reliant and independent from SNAP feel devalued by society and the government. SNAP participants generally report feeling judged by elected leaders, grocery store clerks, SNAP administrators, and the press [21]. Many participants report issues with caseworker professionalism and communications with the local SNAP administrative agency [8]. Further, low-income families face multiple challenges in achieving economic security and upward mobility [22]. Jobs with low wages often have unpredictable and varying schedules [23]. As a result, fluctuations in household income, also known as income volatility, cause low-income families to cycle in and out of eligibility for food assistance benefits [24,25]. Income volatility has increased over the last three decades, while costs for basic needs have continued to increase [13,26]. 

Research has found that income constrained households experience within-month variability in health and educational outcomes, likely due to the instability of recipients’ food access and nutrition throughout the month. For example, one study found that, among low-income individuals, the risk for hospital admission due to hypoglycemia increased by 27% in the final week of the SNAP benefit month [27]. Differences in student test scores have been linked to the timing of SNAP benefit transfers; when exams are taken more than 17 days since SNAP benefits were received, low-income children’s school performance is poorer, which contributes to the academic achievement gap between children of higher versus lower income families [28]. Splitting monthly benefits into multiple allotments has been discussed as a potential policy solution to disrupt the monthly SNAP benefit cycle of adequacy followed by inadequacy. This proposal, however, assumes that the SNAP benefit cycle is a timing problem in which the benefit levels are sufficient, rather than a problem of inadequate benefit amounts [29]. Some suggest that this program modification may have negative effects on rural populations who rely on infrequent grocery shopping trips to minimize transportation costs [30]. One study found that, when the American Recovery and Reinvestment Act of 2009 temporarily increased benefits to all SNAP households, caloric intake did not fluctuate across the month, suggesting that the feast or famine rhythm of the current SNAP benefit cycle may be more related to the amount of benefits than the timing of issuance [31]. 

Given SNAP’s role in reducing food insecurity and what is known about the regular strain on household budgets under normal SNAP issuance patterns, the government shutdown (22 December 2018–25 January 2019) and resulting disruption in SNAP benefit issuance created cause for concern. The shutdown resulted from Congress and the President being unable to agree upon appropriations bills or a continuing resolution (CR) to fund the operations of the federal government [32]. In addition to not being able to pay government employees, many government programs, including SNAP, were unfunded [33,34]. Fortunately, a mechanism for disbursing SNAP benefits during the shutdown was identified. The U.S. Department of Agriculture (USDA) relied on a provision of the just-expired CR, which provided an appropriation for programs such as SNAP to incur obligations for program operations within 30 days of the CR’s expiration [35,36,37]. Despite the operational challenges of this approach, state and county SNAP agencies worked to mitigate harm to clients and issue February 2019 benefits early (within the 30-day period of the former CR expiring) to protect February benefit issuance despite the shutdown. Although participants ultimately did not miss a month of SNAP benefits, they did experience an unprecedented disruption in benefit issuance with a longer than normal gap between SNAP benefit issuance, from a usual maximum of 31 days to up to 50 days. In California, February benefits were issued between January 16 and 20, and March benefits were issued on March 1, a gap of 40–44 days [38,39]. At the time USDA released communications about February SNAP benefit issuance, they were unable to address what would happen in March if the shutdown continued [35,40,41], leaving participants uncertain about their future food access. 

The aim of this study was to begin to capture some of the ways in which the SNAP disruption affected the food security, health, and wellbeing of a small sample of California SNAP participants. Initial questions aimed to capture the baseline (prior to the government shutdown) food security, health, and wellbeing of this convenience sample of SNAP participants. Further questions then explored whether participants’ experiences changed as a result of the shutdown and SNAP benefit disruption. Although the monthly cycle of food insecurity among SNAP participants is well-documented, this study is believed to be the first to capture the short- and potential long-term effects a disruption of SNAP benefits had on participating low-income households.

## 2. Materials and Methods

The Nutrition Policy Institute in partnership with University of California Cooperative Extension Advisors in three California counties received a rapid-response Opportunity Grant from the University of California, Division of Agriculture and Natural Resources, to conduct a cross-sectional qualitative study to capture the experiences of SNAP (called CalFresh in California) participants during the benefit disruption. We collected data in four focus groups, three in English and one in Spanish, with 26 low-income adults in four counties in California from February 28, 2019 to March 15, 2019 (Table 1) during and just after participants experienced a longer than usual gap between SNAP benefit issuance. Counties were selected to include urban (Los Angeles and San Francisco), suburban (San Mateo), and rural (Tuolumne) areas (Table 1). 

### 2.1. Focus Groups

A semi-structured focus group interview guide was developed by the research team and reviewed by three external national SNAP research and policy experts. The guide asked about participants’ usual food routines, the level of support SNAP provides, how they understood and experienced receiving February’s SNAP benefit in January, whether the disruption in benefits impacted their family’s health or stress levels, whether the experience changed the way they think about SNAP, and what they recommend decision makers could do to better help them (Table 2). 

Participating counties were selected to represent some of California’s diversity, including people living in urban and rural areas, speaking primarily English or Spanish, and including different age and racial/ethnic groups. County Cooperative Extension Advisors reached out to community partners in their counties that have strong relationships with SNAP participants and were willing to host focus groups. Participating organizations included a homeless shelter, a middle school, and two food banks (one of which secured an alternate location for the focus group meeting). Agencies were provided with participant recruitment scripts and eligibility criteria (which focused on SNAP participation) and were asked to reach out to a convenience sample of clients to invite to attend the groups. Whenever possible, agency staff called participants with a reminder the day prior to the focus group meeting. 

Two researchers led focus groups: one conducted three groups in English and the second conducted one group in Spanish. Three Cooperative Extension Advisors co-facilitated the focus groups conducted in their counties of work. Focus group participants completed a demographic survey that included questions about gender, age, race/ethnicity, education, employment, income, marital status, food assistance program participation, and food security. Each focus group lasted approximately 1.5 hours and was audio recorded.

### 2.2. Data Analysis

A commercial service was used to transcribe and translate focus group recordings. Four researchers, including a lead researcher and the three co-facilitators, reviewed the transcripts. A codebook was developed based upon the main topics in the focus group guide and was agreed upon by the research team. The lead researcher read and coded all four transcripts and created a summary table of coding results. The co-facilitator of each focus group reviewed the transcripts against the coding results to ensure consistency of interpretations. The researchers used memos to capture themes and emergent findings throughout the analysis process. The research team met four times for 1–2 h each between March and June 2019, to discuss and reach consensus on the findings. 

## 3. Results

Twenty-six adults participated in the study. Most participants were female, ages 31–50, white or Latinx, and reported experiencing an indicator of food insecurity (Table 3). Findings emerged across five themes: (1) the usual struggles participants face in securing an adequate diet; (2) general challenges participants experience utilizing the SNAP program; (3) specific challenges participants experienced with SNAP during the 2019 benefit disruption; (4) the negative impact the disruption had on participants’ food security, stress levels, finances, and perceptions of government support; and (5) participants’ recommendations for SNAP moving forward.

### 3.1. Participants Reported Routinely Struggling to Secure an Adequate, Healthy Diet

The challenges participants routinely confront in trying to feed themselves and their families were discussed multiple times in all focus groups. Participants described using numerous coping skills to ensure they do not go hungry, such as shopping at multiple retailers, coupon clipping, freezing food, and using food pantries. Despite their best efforts, most reported routinely running out of money to buy food during the month, and some reported cutting the size of their own and their children’s food portions.

#### 3.1.1. Routine Food Insecurity Is Experienced in the Context of High Costs of Living

Participants reported that the high costs of living, inadequate or low-wage employment, and/or their limited fixed income from other public programs (Social Security, disability, and unemployment) meant that they routinely faced food insecurity. As one participant said:
You know, with our incomes, and especially with the rent is so high and our bills… it gets really stressful trying to be like, ‘Okay, where’s my next meal gonna come from? How am I going to feed my kids’

Another participant described how food insecurity presented challenges due to her struggles with a health condition:
I have to eat healthy with my- my GI problems that I have. I have to eat white plain chicken, you know, fresh vegetables, and they don’t bother me so much… But … you run out of the chicken, or whatever, you run out of vegetables. Then you’ve got a can of raviolis that the food bank gave you. Well, that’s what you eat. And it tears you up. You know it makes you sick, but you gotta eat.

#### 3.1.2. Food Is Too Expensive, Especially Healthy Food

Participants in all focus groups repeatedly talked about how expensive food is. They described searching to find lower prices on food items, buying junk foods and other “cheap” foods to get more for their money, and struggling to buy healthy foods, which they reported to be too expensive. As one said:
Sodas and all that, it’s easy to last longer, but it’s like trying to do that, they last through the end of the month, and healthy food is hard … you’re trying to make your kids eat healthy, but it’s really expensive at the same time….

### 3.2. Participants Reported Feeling Grateful for SNAP, but Also Felt that the Benefits are Inadequate. Many Reported Negative Experiences with the Program

Participants in each focus group expressed gratitude for the SNAP program; however, the sentiment expressed most frequently was that SNAP benefit levels are inadequate to meet participants’ food needs. Additionally, many participants described negative experiences with SNAP, such as challenges with eligibility requirements or benefit formulas, and difficult communications with caseworkers.

#### 3.2.1. SNAP Benefit Levels Are Too Low

The most frequent issue raised about SNAP was that benefit levels are too low to provide food security and support an adequate, healthy diet. As one said:
… what they give us is not enough for one month, that they should try to help us a little more…. They should think about the children because more than anything else, the food you ask for is for them, they should think about what hurts our children.

Many participants felt that SNAP eligibility and benefit calculations did not adequately account for their high costs of living. While this topic arose in all focus groups, it was more prevalent in the three groups conducted in Los Angeles, San Francisco, and San Mateo counties. In these areas, the cost of housing was routinely discussed as a major challenge. Even in the group conducted in a homeless shelter, a participant talked about trying—and failing—to secure SNAP benefits when she and her husband both worked full-time at low wages, despite their inability to afford rent and other expenses which caused them to lose their housing:
It’s really hard out here… I’m pretty sure all of us pay our taxes, you know, we pay our dues. We do it, everything that we needed to be doing, but yet still we get slapped in the face like, ‘No, you can’t qualify for that because of this and that.’ And that makes it even more stressful, you know?

In the rural county, where housing costs are lower but transportation options are limited and communities are geographically dispersed, participants raised the cost of gas as a major barrier to food security. As one rural participant said:
You have to go back and forth and be able to get to different stores and then maybe be able to go back to a different store in order to get the best prices and everything, but then of course, [gasps] ‘That costs gas.’

#### 3.2.2. Participants Described Challenges with SNAP Administration

Participants raised a number of challenges related to SNAP administration—both with program rules and with customer service and communications. Many described these experiences as leaving them feeling undignified, disrespected, or not cared about. For example, some participants referred to the minimum $15 SNAP benefit this way:

Participant 1:
*I think giving $15 a month to anyone is ludicrous. It’s like an insult.*


Participant 2:
*It’s a slap in the face…. It’s disrespectful is what it is.*


A number of participants reported challenging experiences related to SNAP customer service when dealing with issues of eligibility, enrollment, and participation. One participant said:
There are times when you are disappointed when you go to ask for help, because some workers make you feel that the help you are going to ask for is coming out of their paycheck.

Another said:


*But the communications for this welfare office… I have called them and called them and called them and you know, they won’t return any phone calls… they don’t want to talk to me. I don’t know why, but maybe it’s just the way they treat everybody.*


Finally, a number of participants described confusion and lag time related to frequent changes in their household circumstances that impact their benefits. These changes included adding family members, children leaving the household, changes in employment or eligibility for other public programs, and missing a SNAP administrative deadline, for example a recertification of income eligibility. Lag time refers to delays in budget adjustments when participants with hourly jobs, for example, follow stringent rules about reporting income fluctuations and experience a delayed response in adjusting their benefit that does not align with their needs. Participants generally reported that these situations were confusing, that they struggled to reach caseworkers to get their questions answered, and that the result was uncertainty about the level of benefit they would be receiving in any given month. One participant said:


*…I applied for unemployment, but I was never granted, but I applied. Somewhere in the system… it told them that I-I was receiving benefits from unemployment, which I wasn’t… But CalFresh cut it, like majorly, like 90%…*


Participants in two of the four groups discussed challenges related to benefit levels declining when their earned income increases. One said:
… when you go to work and make your check and then they cut your food stamp down and now, you’ve got to spend your cash now. It’s supposed to be for the bills and you ain’t got cash for the bills and it’s just a vicious circle.

Another participant said that the lag time in response to situational changes meant that needs and benefit levels were not always aligned:
When it goes down, they decrease it in so many days, or weeks, or months after you make the amount. So then you go down again in the amount that you’re making and so then it isn’t working out. You don’t have crap when you need it and you have more when you don’t.

These changes in benefit levels due to program administrative policies create further volatility and uncertainty in SNAP recipients’ finances and compromise their food security. 

### 3.3. Participants Reported Confusion Related to the 2019 Benefit Disruption and Unique Challenges with SNAP as A Result

In California, SNAP benefits are distributed by county social services agencies, and each jurisdiction makes its own decision regarding how and what they communicate with SNAP participants. As such, participants in the four focus group counties reported slightly different experiences with official communications related to the 2019 benefit disruption.

#### 3.3.1. Inadequate Communications

In every group, participants reported receiving the additional benefits prior to receiving formal communication from the SNAP administrative agency. In two of the counties, participants said they never received communication from their county agency or caseworker regarding the benefit disruption, but some participants called the agency to find out why they had the extra money. In one of these counties, one participant said that she used a mobile app, available through the county social services agency, and that the app provided information about the disruption. However, another participant in that county had not gotten the information, saying:
Like, I would have rather have them communicate clearly about what they were doing that month, because I never got the message… So I didn’t know.

In the other two counties, at least some participants reported getting a phone call from the county social services agency. These calls reportedly came at least a day or two after the benefits were distributed, and in some cases participants had spent the benefits before receiving the official notification. In some groups, participants reported learning about the disruption from news stories or from social media or word-of-mouth. One said:
I was hearing stuff about government and stuff like that and everything. So I guess that’s why it was getting messed with, federal stuff.

#### 3.3.2. Confusion

Across all groups, confusion about the benefit disruption was discussed repeatedly. In some cases, the disruption occurred while participants were seeking benefit adjustments for other reasons. As one participant said:
…I turned in my thing and told them, ‘Hey, I only got this. I think I should get an increase from my $15.’ And… so I didn’t know about any of this other stuff going on and them doing their added thing for the government. And so I was confused. I was like, ‘Okay is that my-- is that how much they increased it? They just gave me an extra $15. [laughs] What’s going on?’ And then I called … and they explained it, but yeah, that was very confusing to me…

Although it was the exception rather than the rule, in every county, at least one participant reported spending the February benefit before understanding why they had received the additional funds. Various reasons were provided to explain this. For example, in two of the groups, at least one participant reported worrying that the extra benefits received in January would disappear if they were not spent before the end of that month. As one group discussed:

Participant 1:
*I have also heard that about the food stamps um, um, if you don’t use ‘em, you lose ‘em…. So you be afraid to-- and you can’t get anybody to answer the question.*


Participant 2:
*She’s right. They just send you letters threatening you that, but they never do, do it.*


Another participant described the impetus for spending the unexpected benefit quickly this way:
I was worried that I needed to spend it because with the wacky, screwy way everything is going, I didn’t know if they would take it away with the government shut down, so I had to spend the whole thing. But I got stuff that I could freeze…

In three of the groups, participants reported thinking that the additional benefits received in January were an administrative mistake. Participants expressed various, and sometimes mixed emotional responses to this “mistake.” One said, “We thought we had won the lottery.” Another said:
I was scared. I said, ‘The government made a mistake… I got scared, but I was happy, I thought they were wrong, it scared me but made me happy at the same time.

In every focus group, at least one participant thought that the national SNAP program was ending, and that the extra benefits they received were being provided as a final “bonus” to participants. In one county, this feeling was linked to a message the local social services agency circulated to participants, expressing uncertainty about the March benefit distribution. In other cases, the idea that the program was ending came from word-of-mouth or other sources. As one participant said:
The news I heard said maybe there were going to be changes, that they were not going to give benefits anymore, and that was why they paid the month in advance… I thought that there was not going to be any help anymore… the help is over.

Another said:
I thought that maybe they were going to take it away, because I heard rumors that they were about to take the aid away. So, I said, ‘Maybe they gave us the last month because they want to say that there is not going to be more money.’ … I imagined it this way, but many people were saying that they were going to remove the stamps and that they were not giving them anymore.

In each group, at least one participant talked about splurging in January or treating themselves or their children to something a little bit extra or special. In three of the four groups, this topic was raised multiple times. In some cases, the extra spending was due to confusion or incorrect information about the reason for the additional money in their account. As one said:
We splurged in January, so, um, and not realizing that we were not gonna get benefits in…February … You know, in February, um, we just ate less.

Some participants talked about sharing what they thought were extra benefits with family and/or friends. One said:
I’m going to eat, I’m going to eat well, healthy and well. First, I bought meat, to make roast meat for my children that day, and I invited my siblings who live opposite my house. I told them, ‘Let’s eat roast meat. Help me.’

Another said:
So then when they doubled it, I was like, like I said, we took a bunch of friends to take them to Safeway to get them food and we ate good. Um, we ate things that we couldn’t eat, like a steak.

In other cases, participants knew that the extra funds were for February’s benefits, but they still changed their behavior in January. One said:
I did hear in the news, but I’m like, ‘Let me go get what we need.’ But when you go to the market, you grab more to eat, especially when the kids are there, and you have young kids…. and I know it’s bad, but sometimes as mothers, you don’t know how to say ‘no.’ … Especially you grew up with parents that struggled too… So, for my kids, I try to give them what I didn’t have… It was a struggle, that month.

### 3.4. Some Participants Reported Feeling Relieved or Happy When They Received the February Benefit in January, but the Overall Impact of the Disruption was Increased Stress Levels, Poorer Food Security, Disrupted Finances, and Increased Negative Perceptions of Government Support

Across all groups, the overwhelming sentiment was that the disruption in SNAP benefits had a negative impact. Some people referred to this in a general way, saying things like, “I think we pretty much all agree it kind of screwed us” or “it messed people up.” One said, “It was a 180… going from having extra to having too little.” Only a couple of participants, at least one of whom received the minimum $15 monthly distribution, reported that the disruption had little or no impact.

#### 3.4.1. Emotional Stress

The most discussed impact was stress, which was raised repeatedly in three of the four groups. For many participants, the stress was related to uncertainty—uncertainty about how to make their benefits last for longer than usual, uncertainty about whether the program was ending, and uncertainty about why they received the additional funds on their cards. As one said:
I was stressed because I didn’t know what’s going on.

Another said:
When we were already in a state of chaos and concern and worry, do not make it worse by doing things that we don’t know about, don’t understand, don’t get information on.

Although parents of younger children did not indicate that their children were aware of the disruption, one parent of a teen said:
In my case, my daughter, the oldest, already notices more or sees things. She said, ‘Mom, did they advance you the money because they’re going to take away your help?’ Because medical help is also included, she said, ‘Mommy, are you not going to be able to take me to the doctor, to the dentist anymore?’… She was worried.

#### 3.4.2. Initial Relief, Followed by More Stress

In every group, at least one participant talked about receiving February’s benefit in January as an initial relief. Receiving the benefit during a time of the month when participants’ food budget generally has been exhausted meant some people experienced temporary relief from the usual cycle of scarcity. One participant described the relief this way:
You don’t even think about it, it’s just a big stress relief… That’s a big burden off your back kinda type of thing, how you gonna survive this month literally by eating is, yeah, that’s-that’s something you don’t have to worry about at that point…

Another said:
I was happy because I had a lot of money [laughs] for food.

A few participants talked about being able to eat healthier in January. One said:
I ate a lot better in January, because I had more. I’ve been able to go more to get more fresh vegetables….

However, the relief quickly turned to stress. One participant described it this way:
It’s nice to have more benefits, but if you think about it, you’re going to spend them, and you still have the whole month of February, and say, ‘Okay, I spent them. But what about February? What am I going to do? What’s going to happen then?’

Another described the combined relief and concern this way:
I was able to get a little bit more with that double benefit. Um, I wasn’t limited to and trying to make it stretch for that month. I was able to, like, buy stuff to make a complete meal, you know what I mean? And so it was good. You know, it was good and bad…

#### 3.4.3. Increased Food Insecurity and Negative Financial Impacts

Across all groups, participants talked about negative food security impacts of the benefit disruption, and some talked about negative financial and health impacts as well. One mother talked about utilizing a new charitable food program in February, after she had run out of money to buy food for her family. She said:
The food boxes definitely came in handy, um, um, very appreciative for those. Um, different pantries that give out the eggs, rice and things like that because that’s what they [the kids] love.

Another said:
Well, my daughter [also a SNAP participant]… feeds five people and, um, she said it was really hard for her, because she got all that money in January and… she bought a lot of extra stuff, and she just didn’t have anything much for February…. She stocked up the best she could, but feeding five people and a teenager was very difficult.

Further, the disruption highlighted the economic uncertainty that SNAP recipients live with. One participant described the impact this way:
I felt the impact and it just took me back to feeling poor.

Participants described challenges related to financial impacts of the disruption, because most reported spending their SNAP benefits earlier in the month and running out of money to buy food in February in a way that was different from usual. One said:
That’s what changed this month. In these last two weeks, I had to take from the money we were saving to pay the rent, which had never happened.

For many participants, the disruption to their SNAP benefits led them to have to adjust their finances in ways that meant they continued to have to deal with the after-effects even after the shutdown itself had passed:
She [speaker’s daughter] had to use her gas money for food, because she’s still kind of playing catch up…. She called me several times crying, ‘Ma, I don’t- we don’t have enough food. What am I going to do…? You know, I can’t afford to this and this and this.’ And I can’t help her. So there was a lot of times that all she had was like crackers and whatever. So it was very difficult for them. And I can imagine people with families have the same problem.

Another said:
It’s a huge domino effect, really. It really, really is, ‘cause when you-- You’re just trying to catch up from… February, and here’s March and you just have to get extra stuff that you couldn’t buy in February, you know, and it’s this big old domino effect.

A couple of participants talked about going into debt in February in order to have money to buy food. One said:
Right now, the only difficult decision we have is the stress of paying off what we are borrowing with interest… Having to use it to pay off the rent. You have to do one thing and fail at another. You get into debt with the cards and then it’s a mess.

#### 3.4.4. Reduced Security about Safety net Support and Overall Faith in Government

Participants reported that the benefit disruption impacted their perceptions of the government and overall feelings of security related to government programs. As one said:
You cannot always depend on it [SNAP]. ‘Cause, like he said, we didn’t expect nothing like this happened. So you can kind of, like, never know.

Another said:
I mean, I’m grateful for the government, you know, for it to help me with the food stamps or whatever. But at the same time, I feel like it’s just like they can-- all of a sudden within a blink of an eye, they can take it all away. And then what are we going to do?

In another group, a pair discussed the new uncertainty:

Participant 1:
*And it’s still stressful, because I’m thinking in my head, you know, like, ‘Okay, this is three months solid now that you’ve been playing around with the money for these people. What is it going to be? What is-- is April going to be nothing?’*


Participant 2:
*Yeah, that was my major concern. And that- that is very stressful.*


One participant characterized the increased insecurity of public assistance and the stress it caused this way:
Cause’ the government shutdown… it makes even more things harder and it makes us to think like, ‘Okay, if they did that, you know, what if they do- what if they decide to do that again?’ And then what? Thousands of families going to be left with, you know, with nothing… it just makes us even more and more like what else is… going to happen? … it’s basically, we just feel like it’s a waiting game….That’s just gonna make us worry.

Additionally, in three of the four focus groups, the sentiment that the benefit disruption shook participants’ overall faith in government was raised multiple times. This topic was not discussed in the focus group conducted in Spanish in LA. It was discussed more frequently and with more emotion in the focus group conducted in rural Tuolumne County. One participant said:
It’s obviously caused a lot of… confusion to people and-and the mass hysteria is pointing to, you know, that our government in America can’t get your blank [sic] together, to where we’re all being in confusion and craziness like this. It’s not healthy for us, period. And they just kind of like-like it’s nothing on us, like a game or something to them. This isn’t a game. This is reality.

In another exchange, participants said:

Participant 1:Participant 1: *It’s just disgusting to think that the gover- you know, the government or the powers that be like to have their reason to want that to happen to their people. I mean, I don’t understand what the point of doing that to people is. To see if they can take it or not or what, what’s the deal here?*

Participant 1:Participant 2: *They don’t care, they’re getting their paycheck. They don’t need food stamps. They get paid whether they work or not…*

### 3.5. Participants’ Recommendations for SNAP

Participants expressed a number of ways in which they felt the SNAP program could be improved to better meet their needs and support their families. The recommendations focused on four main issues:Improve benefit adequacy by increasing benefit levels.Modify eligibility and benefit formulas to better address high costs-of-living as well as the expenses associated with working (e.g., transportation, child care).Improve customer service and communications.Do not disrupt SNAP benefits in the future.

The most salient recommendation was to increase the SNAP benefit level. There was strong agreement across participants in all groups that they would benefit from increased SNAP benefits. When asked what additional benefit amount would meet their needs, responses varied. Generally, participants receiving lower benefit levels suggested at least doubling their monthly allotment, such that some participants said an extra $30/month would really help. Participants receiving higher benefit levels suggested larger increases, generally ranging $100–200/month. In two of the groups, the recommendation that SNAP benefits be allowed to be spent on hot foods and/or household items, such as cleaning products and toiletries, was raised multiple times. Sometimes participants suggested providing this in the context of increasing their unrestricted cash benefits. One participant said:
I would, um, actually ask for more cash too, like she was saying. Um, if we could, um, be able to use that EBT [SNAP] money for toiletries, that would, you know, it would just help me a lot.

Participants recommended that SNAP could improve both eligibility determinations and benefit calculations by altering the formulas to better account for costs of living and working. In the three groups conducted in urban or suburban settings, changing the eligibility and benefit formulas to better account for high rents was a priority. In the rural setting, participants recommended more effectively considering transportation costs, as they reported high gas expenses for getting to work, school, grocery stores, and charitable food sites. Further, participants working in jobs with fluctuating hours recommended that the program develop more responsive mechanisms for adjusting to these changes. Participants experiencing changes in family composition, access to public welfare programs, and other changes also recommended that the program become more agile in order to provide the needed support during the timeframe it would be most beneficial.

In the rural focus group, participants agreed that they would prefer SNAP benefits to always be distributed to all participants on the first of the month, as they were in March 2019. Participants said that because they receive cash on the first of the month—either through paid employment, disability, or social security—it is challenging not to receive SNAP benefits at that time. As one said:
… when I get my [non-SNAP] money, I go to Walmart and buy the stuff I need that I have to pay for and I like to buy some of the food there, but then I have to go back out when I get it [SNAP benefits] on the sixth and go to Grocery Outlet and then back to Walmart to buy the less expensive stuff. So it’s-it’s a matter of convenience, but also the cost of gas, time, energy…

Participants in some of the focus groups specifically talked about improving SNAP customer service, especially improving communications with participants. In one group, participants recommended improved communications if a government shutdown or other disruption were to happen again, saying:
I would like to see more literature if that happens again, so that people don’t go overspending and they have enough.

Participants in another group recommended that a wider variety of communication channels be utilized in the event of a future disruption, such as US mail, phone, email, and social media. 

## 4. Discussion

Participants’ stories highlighted their experiences of routine deprivation and struggle, experiences that for some were initially relieved by the disruption when they received February benefits in January, but for nearly all were ultimately exacerbated by it. The most salient themes to emerge were related to routine food insecurity, challenges of affording an adequate and healthy diet, and the inadequacy of SNAP benefits to meet participants’ food needs, all of which were intensified by the shutdown. Despite sharing multiple stories of acting with human agency to cope with difficult life circumstances, as has been reported in earlier studies [9,44], many participants described experiences of feeling vulnerable and barely making ends meet. Participants repeatedly expressed that the SNAP benefits they receive are not enough to provide food security or to enable them to eat an adequate healthy diet throughout the month. Participants described eating cheap foods, foods they do not necessarily like but can afford, and not eating meats, vegetables, and other foods because they are too expensive. Prior qualitative studies also have reported that SNAP participants routinely are unable to meet their food needs with the resources available to them, and that participants perceive healthy foods to be unaffordable [8,9].

In some cases, participants described feeling relief prior to realizing the additional benefit money was an early payment of February’s benefit, amplifying the vulnerabilities inherent in SNAP and creating incremental stress on an already stressed population. Interestingly, some participants said that, even though they knew the additional benefits were for February, having extra money for food at the end of the month in January—when they are accustomed to being unable to buy enough food—provided temporary relief despite knowing it would cause future hardship. Some participants said they were able to eat healthier; parents reported feeling that they could treat their children to favorite foods; others invited friends to share a meal. These stories highlight the basic social feeding experiences that SNAP participants are denied on a regular basis. Having what felt like extra money in January enabled participants to do what many Americans likely take for granted: treating themselves or their children, sharing food with others, or eating the healthy foods that allow them to feel well. 

The challenge the government shutdown presented to social services agencies was not within the scope of this study, but must be acknowledged. That participants received SNAP benefits for February at all can, in many ways, be seen as a tremendous success and social services agencies deserve much credit for scrambling to ensure SNAP funds were distributed. However, it is critical for decision makers, service providers, and the public to understand that the 2019 federal government shutdown and subsequent SNAP benefit disruption felt devastating to many SNAP participants. In California, while some social services agencies communicated with participants via mobile apps, email, or phone messages, many of the study participants did not receive any communication about the benefit disruption from their agency. Even among those who did report receiving agency communications, many said they came after the benefits were received. While we did not probe in depth about how participants would like to receive communications in the future, we did hear that participants have different levels of access to communications technologies. While some relied on landline telephones or network TV news, others were using cellphones, apps, or reading newspapers. While more work is needed to understand local contexts and optimal communications, our work suggests that using multiple channels of communications is likely to be critical due to the variety in access illustrated in our focus groups. 

Most participants talked about this period being very confusing. Participants described knowing it would be difficult not to spend the benefits too early and feeling worried about how they could “stretch” their money to make it last. Many participants reported feeling like they are at the mercy of the government support, for some because they are disabled, for others because they are retired or lack access to more stable or higher paying work opportunities. These participants expressed fear and anxiety about having safety net support when they need it in the future. The confusion, stress, and ongoing uncertainty that resulted from the shutdown were discussed in all focus groups, and in some with a lot of emotion. Many participants also expressed increased or newfound distrust of the government and lack of confidence in the safety net as a result of the government shutdown and SNAP benefit disruption, and felt that they were pawns in a political game that did not fully consider the consequences to them and their children.

Many participants described financial circumstances that change regularly and felt that SNAP benefits are not able to keep pace with the changes. Studies have found that income volatility is widespread, with more than a third of American households facing annual income spikes and dips [45]. Considering fluctuations in income when creating administrative policies related to budget adjustments for SNAP would help to meet participants’ needs.

A recurring theme in all four focus groups was the desire for the safety net system to treat SNAP recipients with more respect and protect their sense of dignity. Although there is research suggesting that the relationship between Temporary Assistance for Needy Families (TANF) caseworkers and their clients may influence client outcomes, little is known about effects of relationship quality between SNAP caseworkers and their clients [8,46]. Our results emphasize the importance of participants being treated with respect by frontline social services staff and caseworkers when they apply for benefits. Another way participants suggested demonstrating respect is by improving communications to recipients and explaining program changes or administrative procedures in a more timely manner. These findings add to what we know about participant frustrations with SNAP benefits being interrupted, reduced, or cancelled without prior notification [8]. Participants also suggested that respect can be demonstrated by increasing benefit levels. Providing a reliable safety net for families who fall on hard times is another way of showing respect and many studies show that SNAP benefits fall short of what households need to ensure a nutritionally adequate diet [11,12,13,14,15]. While SNAP is intended to supplement a family’s food budget, SNAP is the only mechanism by which many families are able to receive minimum levels of nutrition and reduce food insecurity [12]. 

While most of the findings reported were similar across the four focus groups, a couple of differences were notable. For example, the group in rural northern California reported routine use of charitable organizations and positive experiences with them, while they reported more negative feelings about the government. In contrast, the Latina mothers in Los Angeles did not report accessing the charitable food system, and expressed less mistrust in the government. These differences align with recent studies suggesting that political trust is generally higher among Latinx groups [47,48]. Similarly, challenges related to high costs of housing were raised routinely in the Los Angeles, San Francisco, and San Mateo focus groups, but housing costs were not discussed in the rural Tuolumne County group. There, high costs related to vehicle access and buying gas were discussed frequently, yet these costs were not raised in other groups. It is clear that housing costs differ across geographic location and research supports the importance of adjustments on poverty thresholds for geographic variations in housing costs [2,49,50,51,52].

While participants report engaging in a variety of coping behaviors to fend off food scarcity, many strategies require tradeoffs that often compromise health and may have long-term negative financial repercussions [53]. Prior research has demonstrated that SNAP participants do not completely understand the way in which the SNAP program calculates benefits [8,9]. The benefit disruption and the way in which participants talked about how they budget and spend their benefits also suggests that participants do not understand the household food budget assumptions behind the SNAP program structure. While participants clearly express—and research supports—the need to increase SNAP benefits [11,12], participants deserve to be given information about the program structure. The SNAP-Education program offers food resource management education to a small fraction of SNAP-eligible people each year [54]. Better access to financial literacy education could help some participants better understand how SNAP benefits are calculated, and how they can utilize benefits optimally. Although financial literacy would not be expected to fix the challenge of benefit inadequacy in the absence of increases, given the known inadequacy of available support, it may help some participants minimize the harms and this knowledge could help to empower participants to advocate for change. 

This study has several limitations. It was conducted in response to an unexpected disruption in SNAP benefits caused by a sudden and prolonged government shutdown. As such, the research was conducted quickly in order to capture participants’ experiences during the shutdown disruption. We used a convenience sample of counties and participants in California. While we represented different parts of the state and included different population groups, we missed many others. Many of the findings were consistent across groups, but some experiences or concerns that were raised in only one group would have benefitted from testing among additional participants. In no way did this study capture the experiences of all California SNAP participants; more focus groups with diverse participants would be needed to reach saturation. All researchers reviewed at least two transcripts, but not every researcher on the team reviewed all four transcripts. We met regularly to discuss findings, but because all members of the research team work in public health nutrition, it is possible that our shared training and biases have influenced the results. 

Study results reveal the challenges SNAP participants faced during the 2019 government shutdown, even when benefits were issued early in an attempt to mitigate harm, and shed light on routine struggles of SNAP participants with food insecurity. Participants recommended: (1) improving benefit adequacy by increasing benefit levels; (2) modifying eligibility and benefit formulas to better address high costs-of-living as well as the expenses associated with working; (3) improving customer service and communications; and (4) not disrupting SNAP benefits in the future. 

A reliable social safety net system provides citizens with support they can depend upon when they fall on hard times [55]. Given that more than 35 million Americans rely on public assistance to meet their basic need for food, with participation currently spiking due to the COVID-19 pandemic that began in early 2020, it is an important time to increase benefit levels, as has been suggested in multiple studies. Under the current administration, various regulatory changes have chipped away at the federal antipoverty program support that families rely on [56]. The recent federal adoption of measures to reduce SNAP enrollments—such as the public charge rule and the time limit on SNAP participation of able-bodied adults without dependents—suggest that political concerns about cost and dependency currently outweigh concerns about the nutritional health of the poor [57,58,59,60]. COVID-19 has stalled implementation of some of those policies and highlighted the importance of safety net programs, particularly the urgency to feed people during a health crisis. While federal nutrition programs have contingencies built into them to support response and recovery efforts, the pandemic has worsened conditions for families who have the most difficulty affording adequate food [59,60,61]. Provisions included in the Families First Coronavirus Response Act are providing additional SNAP benefits, but, by authorizing USDA to provide the maximum benefit allotment to all households, the provision will not help the nearly 40% of SNAP households that are already receiving the maximum SNAP benefit [61,62].

Further, policies must be established to protect funding for social safety net programs such as SNAP if future shutdowns occur. The security and health of the nation’s most vulnerable people is too much to risk for political expediency. We are not aware of any other research that captured the experiences of the millions of SNAP participants who were impacted by the 2019 government shutdown and subsequent benefit disruption. Their voices and experiences deserve to be heard and known by all stakeholders involved in making SNAP policy decisions as well as those serving SNAP participants. 

## 5. Conclusions

This study suggests that the federal government shutdown and subsequent disruption in SNAP benefit distribution created short- and longer-term negative impacts to SNAP participants, including exacerbating food insecurity, burdening participants with additional stress and confusion, causing financial challenges, and eroding some participants’ faith in government. The disruption highlighted both the importance and the limitations of SNAP in reducing food insecurity among people facing economic disadvantage. New strategies for supporting individuals and families struggling to make ends meet are needed. SNAP can be an important part of the solution, but revisiting eligibility criteria, benefit levels, and customer service practices, as well as establishing policies to protect program participants from future shocks are needed. Conducting research to test solutions can help ensure fair, equitable, and effective policies are adopted.

## Figures and Tables

**Table 1 nutrients-12-01867-t001:** Focus Group and County Characteristics. Description of focus groups conducted with SNAP participants in California to understand their experiences of the 2019 SNAP benefit disruption.

	Los Angeles	Tuolumne	San Mateo	San Francisco
*Focus Group Characteristics*				
Date	2/28/19	3/1/19	3/13/19	3/15/19
Time of day	Morning	Midday	Evening	Afternoon
Number of participants	9	8	5	4
Language	Spanish	English	English	English
Location	Middle school	Food bank	Homeless family shelter	American Red Cross facility
Population	Mothers of school-age children	Adults	Parents raising children	Adults
*County Characteristics*				
Region	Southern CA	Northern CA	SF Bay Area	SF Bay Area
Urbanicity	Urban	Rural	Suburban	Urban
Population [42]	10,039,107	54,478	766,573	881,549
Predominant racial/ethnic groups [42]	Latinx (49%), White (26%) Asian (15%)	White (80%), Latinx (13%)	White (40%), Asian (30%), Latinx (24%)	White (40%), Asian (40%), Latinx (15%)
Median household income (annual in US dollars) [42]	$64,251	$56,493	$113,776	$104,552
Poverty rate [42,43]	23.0%	6.8%	16.5%	18.2%

UC Cooperative Extension Advisors collaborated with community partners, including two food banks, a middle school, and a homeless shelter, to host focus groups. All focus group participants were at least 18 years old and nearly all were participating in SNAP, although a couple were in the process of enrolling or re-enrolling in the program. Focus group participants received a $40 gift card in appreciation of their contributions. The Institutional Review Board of the University of California, Davis, approved the study.

**Table 2 nutrients-12-01867-t002:** **Focus Group Semi-Structured Interview Guide.** Questions asked during focus groups with SNAP participants in California during the 2019 SNAP benefit disruption.

**1**	Thinking about what you and your family eat in a usual month, can you think of any things you are doing that work especially well or any ways that you would like to change what you eat? What would help you meet your goals?
**2**	In a typical month, how much of your family’s food needs would you say are met by your CalFresh/EBT benefit? Where does the rest come from?
**3**	In a typical month, do you find yourself worrying about running out of food because you do not have enough EBT benefits or other resources? What do you do if this happens?
**4**	What did you think about getting a second benefit payment in January?
**5**	Why do you think you got this benefit?
**6**	Can you think of any ways in which getting the February benefits in January changed how you shopped for food or what you ate in January?
**7**	Can you think of any ways that your family’s health was different in January because of this change in how you got CalFresh/EBT?
**8**	Thinking about stress, how would you say the payment of February CalFresh/EBT benefits in January impacted your stress level? Have any changes in stress affected how well you feel? What about your family members?
**9**	Did you have any days that you missed work or school due to health issues in January? Any hospitalizations? Did your children have any changes in their school attendance, behavior, or achievement during January?
**10**	Can you think of any ways in which the longer period between receiving CalFresh/EBT benefits has changed how you shopped for or got food or what you ate in February?
**11**	Can you think of any ways that your family’s health was different in February?
**12**	Thinking about your stress level, how has increased time between CalFresh benefits impacted your stress level?
**13**	What do you want decision makers to know about your experience with this change in CalFresh/EBT payments?
**14**	Has this change in payments led you to think about any ways you’d like to see CalFresh/EBT change? Has this given you any new ideas about your usual CalFresh/EBT benefit levels?
**15**	How can CalFresh/EBT better help you and your family? If your CalFresh benefits could be increased, how much more per week do you think it would take for you to be able to feed yourself/your family?

**Table 3 nutrients-12-01867-t003:** Demographic Characteristics of Focus Group Participants. Characteristics of SNAP participants in California participating in focus groups to understand their experiences of the 2019 SNAP benefit disruption (*n* = 26).

**Characteristic**	***n***	**(%) ***
**Gender**		
Male	4	(15)
Female	22	(85)
**Age**		
18–30	1	(4)
31–50	16	(62)
51–70	8	(31)
70 or older	1	(4)
**Characteristic**	***n***	**(%) ***
**Race/ethnicity**		
White	11	(42)
Latinx	10	(38)
Native Hawaiian/Pacific Islander	2	(8)
African American	1	(4)
Asian	1	(4)
Other	1	(4)
**Highest Education Level Completed**		
Grade 1–12 (no diploma)	9	(35)
High school diploma (or equivalent)	9	(35)
Vocational certificate	3	(12)
Associate’s degree	1	(4)
Professional degree	1	(4)
Other	3	(12)
**Employment ****		
Stay at home providing unpaid care	13	(52)
Work part-time	3	(12)
Student	1	(4)
Retired/Disabled	6	(24)
Unemployed/Laid off	2	(8)
**Income**		
Less than US$16,000	15	(58)
US$16,000–29,000	5	(19)
US$29,001–37,000	2	(8)
More than US$37,000	1	(4)
Do not know	3	(12)
**Marital Status**		
Married	6	(23)
Widowed	2	(8)
Divorced or Separated	11	(42)
Never Married	4	(15)
Living with a partner	3	(12)
**Current SNAP Participant**	21	(81)
**Food Program participation in the past 6 months**		
WIC	6	(23)
Free or reduced-price lunch or breakfast at school	8	(31)
School food backpack program	3	(12)
Food pantry/food bank	15	(58)
**Food Security** (Ran out of food and did not have money to buy more in the past year)		
Often	10	(38)
Sometimes true	15	(58)
Never true	1	(4)

* Due to rounding, some percentages do not add to 100. ** *n* = 25 instead of 26 for this item.

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
