# Peer review of "Participants’ Experiences of the 2018–2019 Government Shutdown and Subsequent Supplemental Nutrition Assistance Program (SNAP) Benefit Disruption Can Inform Future Policy"

_nutrients, 2020, doi:10.3390/nu12061867_

Round 1
Reviewer 1 Report
This manuscript reported findings of a study aimed at a qualitative assessment of participant experiences of disruptions to SNAP benefits during the 2018-2019 government shutdown. This is an important area of research, not only to inform agency action and policy in the event of future shutdowns, but to help understand consequences of other types of potential benefit disruptions. However, this study has some concerning issues that should be addressed.
1 - The study's stated aim is "to capture the ways in which the SNAP disruption affected the food security, health, and wellbeing of California's SNAP participants." (lines 99-100) However, 3 pages of results (lines 187-311) are devoted to detailing problems with SNAP prior to the disruption, problems that have been well-detailed elsewhere and are properly already explored in the introduction and discussion. Results should be mapped to specific aims.
2 - While it is common and accepted to have small sample sizes in qualitative studies, a convenience sample of N=26 to represent the entire state of California's SNAP participants is a very questionable research design. I understand that there were time constraints and these limitations are owned up to in the discussion, but this doesn't undo the reality that these findings can be in no way taken as representative of the population of CA SNAP recipients as a whole. There is a very real danger that the research missed other important concerns that could affect policy choices and agency implementation in the future. I think this can be rectified by describing the study as exploratory, and recasting the aim to "begin to capture some ways in which the SNAP disruption affected...some of California's SNAP participants, in order to better shape additional policy research and future agency responses." Perhaps between addressing comment #1 and #2, the editor and authors might consider reframing the manuscript as a brief report.
3 - The discussion refers to convenience sampling, but nowhere in the methods could I find the recruitment methodology or evidence that there was purposive recruitment within the convenience sample, although based on the focus group demographics, it seemed that might be the case. Regardless, the recruitment and sampling methodology should be detailed in the methods.
4 - The abstract should clarify that is was four focus groups together totaling 26 participants, since the way it reads it seems it might be four focus groups of 26 participants each.
5 - Table 1 should shift the group characteristics headings to the left for clarity (ie "Focus Group Characteristics" and "County Characteristics" should be left-justified). They are too easy to miss for the casual reader.
6 - It is unclear why quotes are reported for most results sections but not others (e.g. 3.2). Also, unique, numeric identifiers should be used for quote attribution, to ensure diversity of participant representation in quotes (standard practice in qualitative research). For example "Quote...." [P04]
7 - The discussion should better differentiate response of CA state agencies versus state agencies' responses in other states. (p14)
Reviewer 2 Report
Thank you for the opportunity to read and review this important work. The authors did a fine job reviewing salient points regarding SNAP and the lived experiences of SNAP recipients during service disruption. Methods and design were clearly expressed as was a description of the sample. I respectfully offer some items for your consideration.
In the discussion, lines 594 - 613 comprise a lengthy paragraph. I might suggest creating a new paragraph starting with line 603 (new paragraph... 'Most participanants ....).
In the discussion section, I'd suggest noting that the study highlighted how disruption of services amplified the vulnerabilities inherent in the SNAP program AND created incremental struess on an already stressed population.
While you note that many received no communications about the changes in SNAP provisions and indicate that communications about program changes are imperative for recipients, it would be helpful to note what channels recipients would prefer to receive this information. (While you did not explore this topic, noting that this would be important to pursue to address this concern, would be helpful).
Section 619 - 632 seems a bit out of place. Previous and following paragraphs are describing participant results; this section is discussing needs and concerns about social safety net programs. I'd suggest placing this section towards the end of the discussion (ie perhaps before the last paragraph).
On line 186, if I understand correctly, I believe the statement shouldl be *n = 25 instead of 26 for this item (25 and 26 got switched around).
On line 660, it appears that a comma is at the end of the sentence (vs a period.)
Round 2
Reviewer 1 Report
The authors have sufficiently address the concerns.